# Reinforcement Compiler Fuzzing

## Abstract

To enforce the correctness of compilers is important for every computing system. Fuzzing is an efficient way to find security vulnerabilities by repeatedly testing programs with randomly modified input data. However, in the context of compilers, fuzzing is challenging because the inputs are pieces of codes which are supposed to be both syntactically and semantically valid to pass front-end checks. Moreover, the fuzzed inputs should be distinct to trigger abnormal crashes, memory leaks or failing assertions that not being detected before. In this paper, we proposed an automatic code synthesis framework called FUZZBOOST based on reinforcement learning. By adopting testing coverage information collected from runtime traces as the reward, we propose a learning system with the state-of-the-art deep $Q$-learning algorithm that optimizes this reward. In this way, the fuzzing agent learns the actions to perform to fuzz a seed program that achieves an overall goal of testing coverage improvement. We have implemented this new approach and preliminary evidence shows that reinforcement fuzzing can outperform baseline random fuzzing on production compilers (i.e. GCC).

## 1. Introduction

Fuzzing is an effective way to find security vulnerabilities in compilers by repeatedly testing the codes with randomly modified inputs. Many existing vulnerabilities are reported by fuzzing techniques (Rash, 2019). Due to the unlimited search space and limited computing resource, existing fuzzing tools explore different methods in fuzz program inputs, that is source code for the scenario of compiler testing, but none can exhaustively examine the entire input space in practice, neither for searching the entire execution paths

[1]Anonymous Institution, Anonymous City, Anonymous Region, Anonymous Country. Correspondence to: Anonymous Author <anon.email@domain.com>.

Preliminary work. Under review by the International Conference on Machine Learning (ICML). Do not distribute.

in target compilers. Therefore, they typically use fuzzing heuristics to prioritize what fuzzing strategies to be taken. Such heuristics may be purely random, or trying to maximize a specific goal, such as code coverage (Kifetew et al., 2014), execution timeouts, errors, crashes (You et al., 2019), etc.

Coverage-guided testing is widely adopted by fuzzers (Zalewski, 2015; Gan et al., 2018; Wang et al., 2018), which utilizes code coverage as the heuristic for search a good next fuzz action from a predefined list. These exhaustive bounded search using domain-specific heuristics and are thereby limited in applicability and scalability. Additionally, they do not benefit from *past experiences* where common knowledge in boosting the fuzzing process across different seeds are shared when similar patterns in the seed files exist. Moreover, most coverage-guided frameworks calculate the rewards/fitness after a single mutation being taken but which overlooks the power of mutation combinations. State-of-the-art methods like AFL (American Fuzzing Lop) (Zalewski, 2017) incrementally add newly fuzzed programs into the seed set according to defined heuristics after each mutation. However, for coverage-guided fuzzing, testing coverage does not increase in a linear way. In other words, each of these mutations may not improve the testing efficacy incrementally. They can even be rejected by lexical or semantic checks on the early stage of compilation. But a trace of mutations may trigger a giant improvement as it may help more to generate a valid and different program to cover more paths inside compilers.

The design of FUZZBOOST is inspired by the exercising of reinforcement learning. Reinforcement learning is about an agent that interacts with the environment, learning an optimal policy, by trial and error, for sequential decision making problems in a wide range of fields in both natural and social sciences, and engineering (Sutton et al., 1998; Bertsekas & Tsitsiklis, 1995). The integration of reinforcement learning and neural networks has a long history with many successfully deployed applications. Recently, with more rapid pace in deep learning (LeCun et al., 2015), benefiting from large amount data, powerful computation, and mature software architectures, more exciting studies appear that adopts reinforcement learning to solve problems that cannot be solved before.

Theoretically speaking, the problem of compiler fuzzing can be seen as a problem of program synthesis, the goal of which is to cover more paths, trigger more crashes or memory leaks in the compiler's execution trace while compiling such new codes. In this paper, we model the compiler fuzzing as a multi-step decision-making process and formalize it into a reinforcement learning problem. We may see the problem of compiler fuzzing as a learning task with a feedback loop. Initially, the fuzzing agent generates new inputs with little knowledge but random heuristics. We will let the compiler run with each new input and as the environment's feedback, for each program execution trace, we capture runtime information gathered from binary instrumentation for evaluating the quality w.r.t. the heuristic we defined for the current input program. For instance, the quality of the generated input can be measured as the number of unique basic blocks on this trace, etc. By taking this quality feedback into account, we construct an end-to-end learning cycle that the fuzzing agent can learn from. By iterating the learning cycle, the agent will be trained to generate a new input program to fuzz compilers in the most effective and efficient way.

We evaluate FUZZBOOST with seed programs from test suites of production compilers, i.e. GCC (GCC, the GNU Compiler Collection, 2019). We conduct experiments of FUZZBOOST with various configurations of the learning framework, including state size, activation functions, etc. To demonstrate the effectiveness of our framework, we also compare it to a baseline system which applies mutation actions with a uniformly distributed strategy. FUZZBOOST outperforms baseline random fuzzing with a higher coverage improvement on a single seed program. Additionally, to show the generalization of FUZZBOOST on boosting the fuzzing process, we design the experiments with seed programs by $\alpha$-conversion. As a result, our tool has a better performance of scalability with a pre-trained model. That means the fuzzing process will be boosted when we reuse an existing model for new seed programs compared with an untrained model.

In summary, we make the following contributions:

- We formalize compiler fuzzing as a reinforcement learning problem by modeling it as a multi-step decision-making process.

- We propose to use deep Q-learning that learns to choose a trace of high-reward mutation actions for any given seed program input.

- We implement a prototyping tool called FUZZBOOST and conduct analysis on real-world fuzzing jobs. It outperforms baseline random fuzzing in terms of testing efficacy.

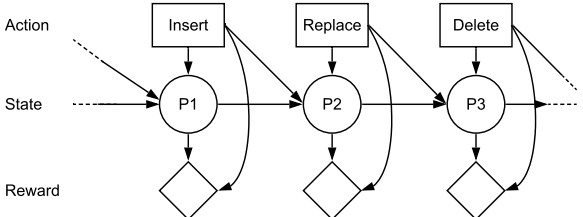

*Figure 1.* Compiler Fuzzing Process

## 2. Overview

Mutation-based fuzzing relies on generating new program inputs by mutating with heuristics based on seed programs. Traditionally, mutation-based fuzzing adopted iterations of one-step fuzzing. In other words, to decide the interest of adding a new mutated input into the seed set, they collect the performance of such input after a single mutation by capturing new crashes in the context of black-box fuzzing or capturing new path information in the context of grey- or white-box fuzzing. However, it overlooks potential performances of a trace of mutations, some intermediate states of which may not be good enough to attract interest or even break the compilation process due to lexical checks on early stages. Therefore, we re-model the problem as multi-step decision-making problem that will give enough attention to these intermediate states being ignored in previous design models. And we formally define the compiler fuzzing and learning process as a Markov decision-making process as described in Figure 1.

As shown in the figure, in this multi-step decision-making process, there is an input mutation engine $M$, that will perform a fuzzing action $a$, and subsequently observe a new state $x$ directly derived from the mutated program $P_2$ by exercising the predicted action $a$ on a original seed program $P_1$. This input mutation engine will predict the program rewrites with regard to an extracted state from the seed program. With the given formalization, it is natural to use Markov decision process (MDP) to model this problem, where the corresponding T-step finite horizon MDP is defined as $M = (s_1, a_1, r_1, s_2, a_2, ..., s_T)$. Here $s_t$, $a_t$, $r_t$ represent the state, action, and reward at time step $t = 1, ..., T - 1$, respectively. To achieve the trace of most effective rewrites of a seed program, our formalization allows us to apply state-of-the-art reinforcement learning methods, in particular, the $Q$-learning (Watkins & Dayan, 1992).

Traditional reinforcement learning is about an agent that interacts with the environment, learning an optimal policy, by trial and error, for sequential decision-making problems in a wide range of fields (Sutton et al., 1998; Bertsekas & Tsitsiklis, 1995). A RL agent interacts with an environment over time. At each time step $t$, the RL agent receives a state

$s_t$ in a state space $S$ and selects an action $a_t$ from an action space $A$, following a policy $\pi(a_t|s_t)$, thus receives a scalar reward $r_t$ and transitions to the next state $s_{t+1}$, according to the environment dynamics, or model, for reward function $R(s,a)$ and the state transition probability $P(s_{t+1}|a_t)$ respectively. In an episodic problem, such as the game of Go, this process continues until the agent reaches a terminal state and then it restarts. The return reward is discounted with a discount factor $\gamma \in (0,1]$, which is written as

$$R_t = \sum_{k=0}^{\infty} \gamma^k r_{t+k}. \tag{1}$$

The goal of the agent is to maximize the expectation of such long term reward from each state. The pick of discount factor usually conforms with the problem design, for which, a value close to 1 is for long time goals but a value close to 0 is more greedy.

### 2.1. Value Function

Value functions are critical for reinforcement learning problems for evaluating a given state at each time step. A value function is a prediction of expected, accumulative, discounted, future reward that evaluates how good each state is, written as $v_\pi(s) = E[R_t|s_t = s]$. It is the expected return for the policy $\pi$ from state $s$. $v_\pi$ decomposes into the Bellman equation:

$$v_\pi(s) = \sum_a \pi(a|s) \sum_{s',r} p(s',r|s,a)[r + \gamma v_\pi(s')]. \tag{2}$$

An optimal state value

$$v_*(s) = \max_\pi v_\pi(s) = \max_a q_{\pi^*}(s,a) \tag{3}$$

is the maximum state value achievable by any policy for state $s$. And $v_*(s)$ decomposes into the Bellman equation:

$$v_*(s) = \max_a \sum_{s',r} p(s',r|s,a)[r + \gamma v_*(s')]. \tag{4}$$

The action value $q_\pi(s,a) = E[R_t|s_t = s, a_t = a]$ is the expected return for selecting action $a$ in state $s$ and then following policy $\pi$. $q_\pi(s,a)$ decomposes into the Bellman equation:

$$q_\pi(s) = \sum_{s',r} p(s',r|s,a)[r + \gamma \sum_{a'} \pi(a'|s')q_\pi(s',a')]. \tag{5}$$

An optimal action value function

$$q_*(s,a) = \max_\pi q_\pi(s,a) \tag{6}$$

is the maximum action value achievable by any policy for state $s$ and action $a$. And $q_*(s,a)$ decomposes into the Bellman equation:

$$q_*(s,a) = \sum_{s',r} p'(s',r|s,a)[r + \gamma \max_{a'} q_*(s',a')]. \tag{7}$$

We denote an optimal policy by $\pi^*$ and this optimal policy is what we want to acquire for general prediction.

### 2.2. Temporal Difference Learning

When an RL problem satisfies the Markov property, i.e., the future depends only on the current state and action, but not on the past, it is formulated as a Markov Decision Process (MDP), defined by the 5-tuple $(S, A, P, R, \gamma)$. We may see the formulated MDP abstraction of our problem in Figure **??**. To solve such problems, when system models are available, we can use dynamic programming methods: that is to use policy evaluation to calculate value function for a policy, and value iteration and policy iteration for finding an optimal policy. However, when there is no model to follow, such as the problem of compiler fuzzing, that no general equations are available to evaluate how good a program is in terms of the testing efficacy of compilers. In this case, we resort to Reinforcement Learning methods and Temporal difference (TD) learning is central in RL. TD learning usually refers to the learning methods for value function evaluation.

TD learning learns value function $V(s)$ directly from experience with TD error, with bootstrapping, in a model-free, online, and fully incremental way. TD learning is a prediction problem. The update rule is

$$V(s) \leftarrow V(s) + \alpha[r + \gamma V(s') - V(s)], \tag{8}$$

where $\alpha$ is a learning rate, and $[r + \gamma V(s') - V(s)]$ is called the error.

$Q$-learning (Watkins, 1989; Watkins & Dayan, 1992) introduces by Watkins, is also regarded as temporal difference learning and was recently combined with deep neural networks (Mnih et al., 2013; 2015) to efficiently learn policies on more complex problems over a larger state space. $Q$-learning is more often adopted in reinforcement learning problems where we choose an off-policy control method to find the optimal policy. $Q$-learning learns action value function, with the update rule,

$$Q(s,a) \leftarrow Q(s,a) + \alpha[r + \gamma \max_{a'} Q(s',a') - Q(s,a)]. \tag{9}$$

$Q$-learning refines the policy greedily with respect to action values by the max operator. Our framework utilizes the deep $Q$-learning which adopts the deep neural network for the $Q$ function. The algorithm for deep $Q$ learning is presented in Algorithm 1. The $Q$-network is initialized arbitrarily with random weights at the beginning. During each episode, we use an incrementally trained $Q$-network for predicting actions in program mutations and retrain the model when we get new rewards for each program state after performing the predicted action. We provide more detailed learning process description in Section 4.

---

**Algorithm 1** Reinforcement Compiler Fuzzing

---

**Output:** action value function $Q$-network
initialize $Q$-network arbitrarily, randomly assign the weights
**for** for each episode $e$ do **do**
    extract state $s$ from seed program
    **repeat**
        $a \leftarrow$ action for $s$ derived by $Q$-network, e.g., $\epsilon$-greedy
        take action $a$, $s'$
        calculate $r$ from runtime trace
        $Q(s,a) \leftarrow Q(s,a) + \alpha[r + \gamma \max_{a'} Q(s',a') - Q(s,a)$
        $s \leftarrow s'$
    **until** state $s$ is a terminal state
**end for**

---

## 3. Design

As described in Section 2.2, we obtain deep reinforcement learning (deep RL) methods for the compiler fuzzing problem specifically when we use deep neural networks to approximate any of the following component of reinforcement learning: value function $q(s, a : \theta)$, policy $\pi(a|s; \theta)$, and model (state transition function and reward function). Here, the parameters $\theta$ are the weights in deep neural networks. We utilize stochastic gradient descent to update weight parameters in deep RL. When off-policy, function approximation, in particular, non-linear function approximation, and bootstrapping are combined together, instability and divergence may occur. However, recent work like Deep Q-Network (Mnih et al., 2015) and AlphaGo (Silver et al., 2016) stabilized the learning and achieved outstanding results. In this section, we will detail the abstraction of the fuzzing process and elaborate on how to map this process into a deep reinforcement learning process.

In reinforcement learning, one episode is one complete sequence of states, actions and rewards, which starts with an initial configuration and ends with a terminal state. For example, playing an entire game can be considered as one episode, the terminal state being reached when one player loses/wins/draws. In the problem of compiler fuzzing, one episode can be defined as generating a good program by mutating an existing seed program (initial state) with respect to the defined quality and in our preliminary implementation, we hard-coded the entire trace length of program mutations as one of the terminal conditions (terminal state).

Before we start the learning process, we first start with a randomly generated neural network. After initiating a new episode, State 0 is initiated by preprocessing a seed program $P$. We initially extract a substring within this seed program with the window size $w$ and offset $s$. By observing this substring, the neural network will help us to predict a mutation action to be taken. Mutation actions are being taken on token-level which include *insert* a token, *switch* two or more tokens, *replace* a token, or *change* the window size or offset to enable another substring to observe and mutate. Once an action is being taken, we run the compiler (any production compiler) with the program after mutation and calculate the reward $r$ of this new program with a record of the execution trace. With the increased number of actions being taken, we deduct the reward by a discounted rate $\gamma$ which is a value between $0$ and $1$. The state will move to State 1 after one action being taken. We iterate the mutation prediction and evaluation until a *terminal* state. There are four key elements in this process: *action*, *state*, *environment*, and *reward*. We will elaborate on these key elements one by one.

### 3.1. State

A state $S$ is a concrete and immediate situation in which the agent finds itself; i.e. a specific place and moment, an instantaneous configuration that puts the agent in relation to other significant things. It can the current situation returned by the environment, or any future situation. In the problem of compiler fuzzing, the agent learns to interact with a given seed program. Therefore, the state is a function about a given input seed program $p$. In our design, the interaction is performed upon the observation of substrings of consecutive symbols within such an input. Formally, let $\Sigma$ denotes a finite set of symbols. The set of possible program inputs $I$ in this language is defined by the Kleen closure $I := \Sigma*$. For an input program string $p = (p_1, p_2, ..., p_n) \in I$, let

$$S(p) := (p_{1+i}, p_{2+i}, ..., p_{m+i})|i \geq 0, m + i \leq n \quad (10)$$

denote the set of all substrings of $p$. We define the states of the Markov decision process to be $I$ and $I$ is a union set of $S(p)$. Thus, we have $p \in I$ denotes an input program and $p_0 \in S(p) \subset I$ is a substring of this input seed program. The entire state space of a seed program is $S(p)$, which is theoretically infinite since any symbol in this language $I$ can be involved after mutation. In other words, the seed program can be converted into any other programs that conforms the programming language grammar.

### 3.2. Action

Action $A$ is the set of all possible mutations the agent can perform. An action is almost self-explanatory, but it should be noted that agents choose among a list of possible actions. In the problem of compiler fuzzing, we define the set of possible action $A$ of our Markov decision process to be mappings of extracted substrings $S(p)_0$ to probabilistic rewrite rules. The rewrite rules are defined in accord with the extracted substring and predicted type. In a high-level, we define two types of rewrites, on the extracted content

and on the extraction window. To be more specific, the rewrites of extracted content are performed on token-level which include *insertion*, *replacement*, *re-ordering*, *deletion* and *replication*. These pre-defined token-level rewrite rules conform with C language lexical requirements. The neural network will predict which type and on which position an action should be performed and we employ a lexical analysis on such extracted substring to conduct such mutations on a finer-grained granularity. This will change the input program $p_i$ into $p_{i+1}$ by mutating the substring $S(p)$ in observation, and meanwhile, keep the original syntactic and semantic validity with the best effort. For the second type mutations, they are designed to make a change of extraction windows. The atomic mutations include window *left shift* and *right shift*; and window size *up* and *down*, one character length for each. Each of these actions does not modify the original seed program but motivates an originally extracted substring $S(p_i)$ into another substring $S(p_{i+1})$. For both types of mutations, the time step increases to next state until termination on the current episode. The substring rewrites will consider every substring in the seed program and predict accordingly to maximize the accumulated rewards along the mutation trace. We also define a *terminate* action to early stop the mutation episode. That is to say, the mutation agent can actively terminate a mutation episode while observing the extracted substring.

### 3.3. Environment

The environment is the world through which the agent moves. The environment takes the agent's current state and action as input, and returns as output the agent's reward and next state. In the problem of compiler fuzzing, the environment is the compiler or verifier. To observe more detailed information about the fuzzing efficacy, we develop a plug-in based on program execution traces. That is to say, we record dynamic traces when running any production compilers, i.e. GCC, with generated programs. In compiler construction, a basic block of an execution trace is defined as a straight-line code sequence with no branches except for the entry and exit point. We capture all the unique basic blocks $B(\mathrm{T}_p)$ with respect to each execution trace $\mathrm{T}_p$, and calculate a store with all the unique basic blocks covered by the existing test suite $I'$ so far. In our implementation FUZZBOOST, the program execution trace is generated by Pin (Luk et al., 2005), a widely-used dynamic binary instrumentation tool. Pin provides infrastructures to intercept and instrument the execution trace of a binary. During execution, Pin will insert the instrumentation code into the original code and recompiles the output with a Just-In-Time (JIT) compiler. We develop a plug-in of Pin to log the executed instructions. Additionally, we develop another coverage analysis tool based on the execution trace to report all the basic block touched so far. It will also report whether and the number of

new basic blocks are covered by a certain new program in the compiler code. Additionally, our environment will also log and report abnormal crashes, memory leaks or failing assertions of compilers with the assistance of internal errors alarms from the compiling messages.

### 3.4. Reward

Rewards provide evaluative feedbacks for an RL agent to make decisions. However, rewards may be sparse so that it is challenging for learning algorithms, e.g., in computer Go, a reward occurs at the end of a game. There are unsupervised ways to harness environmental signals. The reward function is a mathematical formulation for rewards. Reward shaping is to modify the reward function to facilitate learning while maintaining the optimal policy. In the problem of compiler fuzzing, to motivate the testing coverage, we define the reward relates to the unique basic blocks covered by a certain generated program $p$ and the entire test suite $I'$; that is

$$R(p, I') := B(\mathrm{T}_p) / \bigcup_{\rho \in I'} B(\mathrm{T}_\rho), \qquad (11)$$

where $B(\mathrm{T}_p)$ is the unique basic blocks of the execution trace of a program $p$ and $I' \subset I$ is all the programs generated so far in the test suite. This stepwise reward $R$ is a continuous scalar value that has a range of $(0, 1]$, where $1$ is achieved when a specific execution trace covers all the basic blocks that have being tested so far by existing test cases. This reward motivates the mutation steps towards the training purpose: improve the compiler testing coverage by selecting a critical subsequence inside a seed program and making simple mutations in a trace.

## 4. Learning

To start a deep $Q$-learning process for compiler fuzzing, we propose FUZZBOOST which adopts a constructed forward neural network with two layers connected with non-linear activation functions. We build this end-to-end learning framework with the environment reward calculated based on dynamic trace analysis. In this section, we present the overall learning process for FUZZBOOST by illustrating the fuzz action prediction in the reinforcement learning process of compiler fuzzing as shown in Figure 2.

### 4.1. Initialization

We start with an initial input seed $p \in I$, where the choice of $p$ is not constrained but can be any C program even it is not well-formed. We employ the GCC test suite as our sampling pool and randomly selected programs to be our seed inputs. We propose to use a neural network as the $Q$ function to mimic the reasoning for input mutation of compiler fuzzing. This deep neural network maps states

```
foo(a, p) int *p; { p[0] = a;
a = (short)a; return a;} main
() { int i; foobar(i, &i); }
bar(a, b) { int c; c = a % b;
a = a / b; return a + b; }...
```

*Figure 2.* Fuzz Action Prediction in the Reinforcement Learning Process of Compiler Fuzzing

(embedding of an extracted substring from seed programs) to estimate $Q$ values for all actions $A$. Due to the lack of heuristics at the very beginning, we built it a reinforcement learning process, where the neural network is randomly initialized and gradually tunes the parameters $\theta$ with learned mutation heuristics calculated with environment rewards.

### 4.2. State Extraction

FUZZBOOST will observe a substring within a seed program to predict actions to perform. The substring is extracted from the seed program by customized window, and encoded as $State(p)$. In Section 3.1, we defined the states of our Markov decision process to be $I = \Sigma *$. To be more specific, it is a strict substring $p'$ at offset $o \in 0, ..., |p| - |p'|$ and of window size $|p'|$. To make the extracted state controllable, we defined actions in Section 3.2, to shift and resize the window. By performing window-related actions, the fuzzing agent is able to see the whole program via partially observe fragments consecutively. In other words, FUZZBOOST will learn to select the most critical piece of code to mutate incrementally during the training process.

### 4.3. Deep $Q$-Network

We implemented the $Q$-learning module in Tensorflow (Abadi et al., 2016). The deep neural network that used for prediction is a forward neural network with two hidden layers connected with non-linear activation functions. The two hidden layers contain 100 and 512 hidden units respectively, and fully connected with an input layer with 100 units (which is the max window size for input substring) and an output layer with 10 units (which is the size of action space). The goal of the training is to maximize the expected policy reward. Since the MDP is a finite horizon in our practical design, we adopt a discount rate $\gamma = 0.9$ to address the long-term reward. We set the learning rate $\alpha = 0.001$ to achieve our best-tuned results. We use the decayed epsilon-greedy strategy for exploration in the reinforcement learning iteration, that is the $\epsilon$ value was set up to 1 at the very beginning and decays over time until

a min value, $0.01$ in our configuration, is reached. In this scenario, with the probability $1 - \epsilon$, the agent selects an action $a = argmax_{a'}Q(x_t, a_t)$, which is the estimated optimal by the on-training neural network. On the contrary, with probability $\epsilon$, the agent explores any other actions with a uniformly distributed choice within the action space $|A|$. To evaluate the proposed framework with the deep $Q$-Network, we explored its effectiveness under several different initial state sizes. We also explored several non-linear activation functions, including *tanh*, *sigmoid*, *elu*, *softplus*, *softsign*, *relu*. We report experimental results in Section 5.

### 4.4. Termination

A mutation episode will terminate when the agent detects a terminal state. In our design, we define three conditions that may trigger the terminal state of mutating of a single seed program: (1) the agent executes the "terminate" action from the neural network prediction; (2) the generated program reaches a maximum number of mutation steps; or (3) the agent generates an invalid action that triggers miscellaneous effects during the reward calculation. The first type of termination will cut the program mutation actively by FUZZBOOST while the latter two are passively ended with pre-defined policies. We hard-code the limitation of mutation trace length to be 20 in all of our experiments. Theoretically speaking, from the perspective of fuzz testing, the mutation trace can be generated as long as possible to achieve enough randomness. But in practice, to improve the testing efficacy in a most effective way, we set up these policies to enforce our learning agent to learn within the shortest path.

## 5. Experiments

In our research, we proposed a reinforcement learning framework FUZZBOOST that incrementally trains a deep neural network to predict mutation actions on a given seed program that improves the compiler testing coverage in a most effective way. We evaluated FUZZBOOST based on a seed input set gathered from the GCC test suite. We

| State Size | 50 | 60 | 70 | 80 | 90 | 100 |
|---|---|---|---|---|---|---|
| **Coverage Improvement (%)** | 37.14 | 36.11 | 30.29 | 28.95 | 28.07 | 27.94 |

*Table 1.* Coverage improvements with different state size

| Activation Function | tanh | sigmoid | elu | softplus | softsign | relu |
|---|---|---|---|---|---|---|
| **Coverage Improvement (%)** | 37.14 | 28.27 | 7.48 | 13.72 | 14.22 | 13.26 |

*Table 2.* Coverage improvements with different activation functions

randomly sampled 20 C programs in the test suite as our benchmark problems. We evaluated FUZZBOOST in terms of the testing efficacy and scalability. We also addressed the issue that the fuzzing process can be boosted with a pre-trained model even if we reuse for new seed programs. All measurements were performed on i7-7700T 2.90Ghz with 12GB of RAM.

### 5.1. Testing Efficacy

Coverage improvement is the most important measurement for testing. It denotes the overall lines/branches/paths in the original code is being visited. In our design, we use the accumulated number of unique basic blocks being executed with the generated new test cases as an alternative to represent the code coverage. To show that FUZZBOOST learning algorithm learns to perform high-reward actions given a seed input observation, we compare the improved testing efficacy against a baseline random action selection policy. The choice of the baseline method uniformly distributed among the action space $A$ and we terminate the actions with the same methodologies as our method described in Section 4.4. We randomly sampled 20 C programs in the GCC test suite, specifically, from the *gcc.c-torture* repository.

*Baseline:* We performed the experiments with the two different action selection strategies using each of the programs from the sampling pool as the seed. We generated 1,000 new tests from both strategies from the seeds and recorded the accumulated number of unique basic blocks along the execution trace. In general, FUZZBOOST improved the testing coverage by 37.14%. Figure 3 shows the coverage improvement of four comparisons, among which the most and least improvements, 79.17% (case 1, seed1.c) and 12.24% (case 2, seed2.c) respectively, are achieved. To conclude, an improvement of 5.59% coverage improvement is achieved at most with the newly generated 1,000 programs by FUZZ-BOOST for a single seed.

*State Size:* We increased the initial state size $w = |x'|$ from 50 characters to 100 characters and measured the average reward improvement compare with the baseline strategy on seed1.c. Table 1 shows the results for this experiment. We can see a decreasing improvement when increasing the initial state size. To interpret this result, smaller substrings are better processed than larger ones. In other words, our

model learns the best move of small windows and will select the best action accordingly to improve coverage.

*Activation Function:* We are also interested in testing efficacy improvements when applying different activation functions in the proposed model. We conducted experiments to generate 1,000 new program upon seed1.c with FUZZ-BOOST trained with models using different activation functions. Table 2 compares the different activation functions with respect to improvement of coverage. For all activation functions provided by the Tensorflow framework, we found the *tanh* function to yield the best result for our setting.

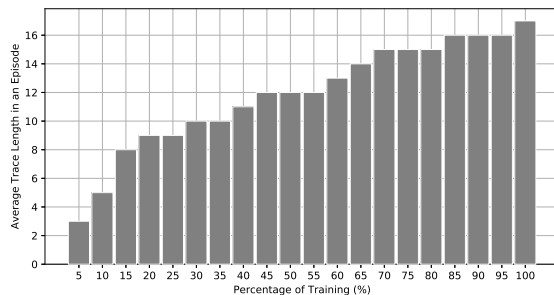

*Figure 4.* Average Mutation Length in an Episode with Training

*End State:* We define the compiler fuzzing as a multi-step decision-making problem and set up the end-to-end learning framework. Theoretically speaking, not like the problem of Go, the end state of FUZZBOOST is not deterministic in all cases. In our design, we hard-coded a limit on the length of mutation traces for experiments, but naturally, the traces can be endless to gain enough randomness and achieve a higher reward. We also designed that the trace can be terminated by the agent itself or triggered by miscellaneous effects during the dynamic analysis. Thus, we are also interested in the distribution of the trace lengths under different configurations. Figure 4 shows the trace length distributions along the learning process. From the result, we can see that, with the training goes on, the trace lengths are increasing. That is to say, the fuzzing agent tends to not to cut off the mutation.

### 5.2. Boosting with pre-training

We next address the question, given an agent which is pre-trained on seed programs $P_{train} = p_i \sim P$, will this agent improve the testing efficacy faster than learning from

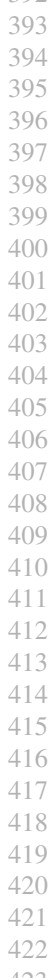
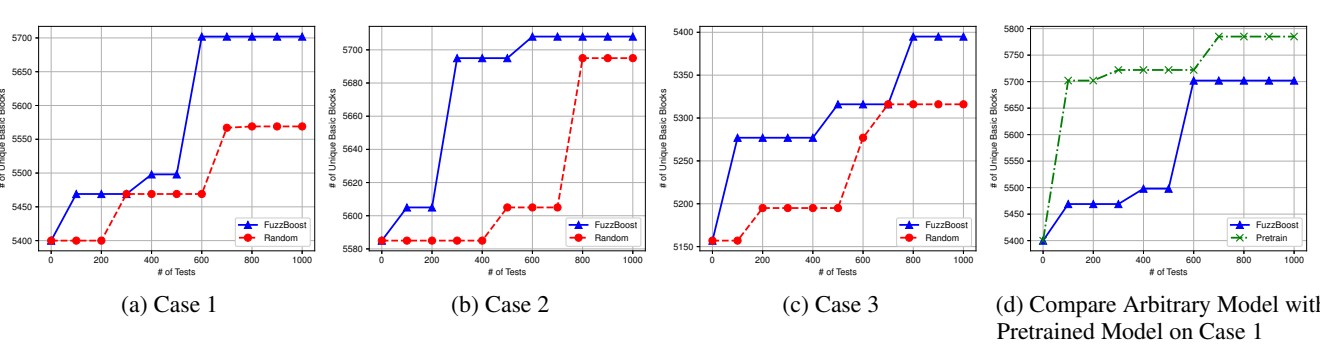

| (a) Case 1 | (b) Case 2 | (c) Case 3 | (d) Compare Arbitrary Model with Pretrained Model on Case 1 |

*Figure 3.* Number of Unique Basic Blocks

scratch? We prepare the training and testing data as follows. We took *case 1* which has the most coverage improvement from the initial 20 seed program and created 9 $\alpha$-equivalent programs for this seed program. We call a program $P'$ is an $\alpha$-equivalent program of program $P$ when we only perform bound variable renaming on $P$. We used 80% of them serves as $P_{train}$ and the rest 20% are used for $P_{test}$. After pre-training the agent on $P_{train}$ for 50 epochs, we saved the model and reused it on $P_{test}$. It continued the trial-and-error reinforcement learning.

Figure 3d shows coverage improvement using FUZZBOOST with an initially arbitrary model and another pre-trained model respectively. We may see that the coverage improvement for the latter case improves drastically towards the highest value in the former case despite the minor difference in the language of two seed programs. In addition, with the training goes on, the coverage was again improved to a new highest value that outperformed previous testing efficacy. It reveals the transferability of a trained model in the context of compiler fuzzing.

# 6. Related Work

Our study is related to the following aspects of research.

## 6.1. Deep Reinforcement Learning

Despite the popularity in solving the game of Go, reinforcement learning is also adopted as a powerful technique for program synthesis. Brunel et al. performed reinforcement learning on top of a supervised model with an objective that explicitly maximizes the likelihood of generating semantically correct programs (Bunel et al., 2018). Researchers also proposed Neurally Directed Program Search (NDPS) (Verma et al., 2018), for solving the challenging non-smooth optimization problem of finding a programmatic policy with maximal reward. Our target is to generate source programs that are well-formed but contain different syntactic features to trigger compiler errors. In our design,

we may consider the improvement of testing coverage into the reward as feedback for reinforcement learning.

## 6.2. Mutation-based Fuzzing

Mutation-based fuzzing uses an existing corpus of seed inputs for fuzzing. It generates new inputs by modifying the provided seeds. A well-known fuzzer that is mutation-based is called AFL (Zalewski, 2017) which randomly mutates seed inputs and incrementally add new seeds into the set with respect to defined heuristics. Several boosting techniques are proposed to improve the efficiency of mutation-based fuzzing. AFLFast (Böhme et al., 2017) boosts up original AFL fuzzer by focusing on low-frequency paths that allow the fuzzer to explore more paths with limited time. Skyfire (Wang et al., 2017) applies the knowledge in existing seed inputs for fuzzing programs that take highly-structured inputs. Kargen and Shahmehri (Kargén & Shahmehri, 2015) perform mutations on the machine code instead of directly on a well-formed input that they can use the information about the input format encoded in the generated program to produce high-coverage inputs. DeepFuzz (Liu et al., 2019) utilized an RNN-based model to encode program grammar and generate new well-formed C programs for compiler fuzzing. In this paper, our method boosts the mutation process by using a deep neural network to predict the mutation based on an observation of existing seed programs.

# 7. Conclusion

In this paper, we proposed FUZZBOOST, a deep reinforcement learning framework to fuzz off-the-shelf compilers by generating new programs with coverage-guided dynamics. Our proposed end-to-end learning framework learns to select the best actions to perform automatically without any supervision. It improved the testing coverage on a seed set from the GCC test suite and outperformed the baseline fuzzing agent with a random selection strategy. Moreover, after being pre-trained, it can generalize the strategy to new instances much faster than starting from scratch.

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
