# OpenReview forum: "Reinforcement Compiler Fuzzing"
_ICML.cc/2019/Workshop/RL4RealLife — Submitted to RL4RealLife 2019_

### Official Review · AnonReviewer1 · 2019-05-26
**Reinforcement Compiler Fuzzing**

**Rating:** 3
**Confidence:** 3

**Review:**

Summary: This paper applied reinforcement learning to mutation-based program synthesis for compiler fuzzing. The mutation process is regarded as MDP and optimized via DQN. Comparing with other mutation-based methods with iterations of one-step fuzzing,  this method can benefit from the past mutation actions, which is helpful to obtain an optimal mutation trace.

strengths:
 (1) The topic is very interesting, and the methods are reasonable.
 (2) Preliminary results have been achieved comparing with random fuzzing method.
 (3) The paper is well-written.

Weaknesses:
(1) A mismatch between introduction and experiments. In the beginning, the author focus on the advantage of their proposed methods over previous heuristic mutation-based fuzzers. However, no comparisons have been made in the experiments.
(2) The random fuzzing baseline is not very strong.

Questions & Comments:
(1) This paper uses the covered unique basic blocks as the reward. I think such a reward cannot motivate the generated program to trigger more crashes or memory leaks. How did you solve this problem?

---

### Decision · Program_Chairs · 2019-05-25

Reject